# Effect of Microstructural and Tribological Behaviors of Sputtered Titanium Carbide Thin Film on Copper Substrate

**DOI:** 10.3390/ma16010174

**Published:** 2022-12-25

**Authors:** Musibau Olalekan Ogunlana, Mammo Muchie, Oluseyi Philip Oladijo, Mutiu Erinosho

**Affiliations:** 1Industrial Engineering Department, Faculty of Engineering and the Built Environment, Tshwane University of Technology, Private Bag X680, Pretoria 0001, South Africa; 2Chemical, Materials and Metallurgy Department, Botswana International University of Science and Technology (BIUST), Private Bag 16, Palapye Plot 10071, Botswana; 3Department of Mechanical and Industrial Engineering, University of Namibia, Ongwediva 15006, Namibia

**Keywords:** copper, magnetron sputtering, microstructure, wear, coefficient of friction, titanium carbide

## Abstract

Titanium carbide (TiC) thin films were deposited by radio frequency magnetron sputtering (RFMS) onto a copper substrate by using Argon (Ar) gas plasma at a gas flow rate of 10.0 sccm. The effect of time and temperature at a constant RF power on the structural and tribological properties were respectively investigated by atomic force spectroscopy (AFM), X–ray Diffraction (XRD), Fourier transform infrared (FTIR) spectroscopy, optical microscopy (OM), scanning electron microscopy (SEM) and tribological measurements. All films were tested to have crystal structures with the preferential plane (111) and dominant plane (200) grain orientations. Thus, plane (111) has phase identification of Cu(Cu16Ti)_0.23_ for some samples, whereas plane (200) has a phase identification of Cu(Cu_0.997_Ti_0.003_) and Cu(Cu_0.923_Ti_0.077_) for other samples. The lowest thin film roughness of 19.797 nm was observed in the sample, with RF power, sputtering time, and a temperature of 200 W, for two hours and 80 °C, respectively. The FTIR spectra of TiC films formed under different sputtering times (2–3 h) and temperatures (80 °C–100 °C) on Cu substrates at a constant sputtering power of 200 W in the range of 5000–500 cm^−1^. The peaks at 540 cm^−1^, 780 cm^−1^, and 1250 cm^−1^ are presented in the FTIR spectra and the formation of a Ti–C bond was observed. On the other hand, a sample was revealed to have the lowest wear volume of 5.1 × 10^−3^ mm^3^ while another sample was obtained with the highest wear volume of 9.3 × 10^−3^ mm^3^.

## 1. Introduction

Transition metal-based carbides, such as titanium-carbide (TiC), are materials with outstanding and superior properties which include surface hardness, good corrosion and wear resistance, low coefficient of friction, and high electrical conductivity [1,2]. TiC coatings are extensively used as materials for wear resistance in several manufacturing applications, such as cutting tools and biomedical materials, which are, however, due to their high thermal and chemical stability. In addition to the mentioned properties, TiC is highly attractive when considered for high-temperature applications, simply due to its chemical inertness, good refractory, and superior modulus properties [1]. On the other hand, wear in the manufacturing and production industries is regarded as a vital hindrance to optimum mechanical performance and efficiency, in which friction is the principal cause of wear and dissipation of energy. Wear could be decreased with the help of wear resistance coatings on the surface of the active part materials [2]. In wear resistance, the film’s lubrication reduces wear and friction force fluctuation [3]. Both tribological and mechanical behaviors of brittle material deposited on a substrate material would actively be influenced by in-service loading application and residual stresses. However, fatigue and failure of hard coatings on the desired substrate in majorly tribological situations are however established through the delamination of coatings from the substrate, whereby it generates a fracture attributed to adhesive and cohesive failures.

Consequently, failures in thin film-coated materials are predominantly related to very high compressive-residual pressure on the coatings [4]. In addition, ceramic materials, such as zirconium nitride (ZrN), titanium nitride (TiN), titanium carbide (TiC), vanadium carbide (VC), tungsten carbide (WC), and tantalum nitride (TaN), have been successfully used as protective coatings due to their outstanding resistance to corrosion, resistance to wear, and superior mechanical, refractory, and electrical properties [2,5,6]. Ceramic thin films, such as TiC, are frequently used in cutting tools to increase the operational life and cutting efficiency of the tools. Thus, TiC material is characterised by relatively low density (4.91 g/cm^3^), higher melting temperature (Tm = 3340 K), superior modulus (410–510 GPa), good thermal stability/conductivity, as well as higher hardness (HV = 28–35 GPa). In addition to the properties exhibited by TiC thin films, it also shows a low oxidation resistance, brittle in nature, and high stress, which adversely hinder its wide range of industrial applications [7]. Hence, the desirable thin film of TiC properties could be obtained via physical vapour deposition (PVD), such as radio frequency magnetron sputtering (RFMS) due to its strong adhesion, ease of control of processing parameters (i.e., deposition rate, sputtering power, working pressure and gas flow rate) [8,9,10,11]. RFMS is a method widely used to deposit thin hydroxyl-apatite (HA) bioactive coatings on metallic substrates with substantial uniform films, which make it suitable for industrial/manufacturing applications [9,10].

Magnetron sputtering (MS) techniques are generally considered to be preferable to produce high-purity and dense films when compared with other PVD techniques; since they can be used when a coating with specific properties is required. Therefore, RFMS is a coating method that is used to monitor and control the effect of coating parameters, such as film thickness, gas flow rate, sputtering temperature, and sputtering power [12,13,14]. Thus, this paper reports analysis regarding preparation, microstructural properties, and tribological behaviour of TiC–Cu thin film deposited by using the RFMS method. Hence, the effects of deposition time and temperature at a constant sputtering power on the surface morphology, structural properties, and the tribological behaviour of TiC thin films on copper substrates are investigated.

For sustaining the life cycle of the base material during manufacturing industries operation, titanium carbide (TiC) thin film is fabricated on copper (Cu) substrates using the radio frequency magnetron sputtering (RFMS) method for surface engineering applications. The RFMS deposition process is a technique used for coating material to achieve certain properties, such as morphology and structural properties (i.e., surface roughness, grain size, and growth smoothness), high hardness, corrosion resistance, and resistance to wear [1,2]. Thus, TiC is a member of ceramic materials, brittle in nature, and coated on metallic copper, which is ductile in nature, in order to achieve improved properties of the parent material. The defect in engineering materials calls for an imperative solution, using an advanced coating method to fabricate a material with substantial properties. Hence, the RFMS technique is used for the deposition process of TiC on Cu materials; because it accommodates uniform and fully dense thin film fabrication. The magnetron sputtering technique is categorized as the PVD method that could require a gas pressure below 1 Pascal (Pa) [15]. However, magnetron sputtering provides a high ionisation efficiency of electrons when compared with a conventional diode or cathode-sputtering system. The magnetron sputtering technique provides a solution to the generation of low ions; which resulted in a high rate of deposition by placing magnets behind the target.

Therefore, the RFMS thin film deposition method is adopted for this study simply because it produces a smooth film of flexible sputtering process parameters with the help of a rotating sample holder for even distribution and fully dense film deposition. However, this method of RFMS and the selection of materials were chosen due to their outstanding properties, such as stable thermal conductivity to prolong the life of the parent material during operation and to fabricate a material with superior properties for cutting tools, automotive, and aerospace applications.

This research work was performed experimentally using microstructural and tribological analyses to investigate the effects of TiC thin film coated on metallic copper plates/substrates for surface manufacturing applications.

## 2. Materials and Methods

### 2.1. Thin Films Deposition

With the RFMS technique, TiC is obtained as the target material for the thin film deposition on the copper substrates with an area of 100 mm × 23 mm and a thickness of 1 mm. Figure 1 shows the schematic illustration of the RFMS technique of TiC thin film coated on metallic copper substrates.

The high purity (99.5%) solid TiC target (HHV Ltd., Company, Crawley, United Kingdom) of 75 mm in diameter and 3 mm thickness, with a copper backing plate of 3 mm, was used. The copper substrates were, however, cleaned with acetone to remove any unwanted particles and contaminated residue from the substrate’s surface. The films were grown using RFMS equipment (HHV–TF500 Ltd., Company, Crawley, UK) with constant power and varying deposition time between 2 and 3 h, whereas the temperature between 80 and 100 °C was also observed. Prior to commencing the deposition process, a 10-minute pre-sputtering was performed to flush any unwanted and contaminated particles out of the target’s surface. The base deposition pressure was recorded to be 5.63 × 10^−2^ Torr, whereas the thin films were deposited at a working pressure of 4.68 × 10^−2^ Torr. The TiC deposition was carried out in the non-reactive environment by using Argon at a gas flow rate of 10.0 sccm. The sputtering chamber was designed to place the target and the substrate at 130 mm distance. In this study, the substrates were placed in rotation holders to achieve uniform film deposition. In addition, Figure 2 shows the prepared photographic image of TiC thin film coated on a metallic copper substrate by using the radio frequency (RF) magnetron sputtering (MS) method. Thus, copper material was selected for this research work due to its outstanding chemical, thermal, and tribological properties. RFMS equipment was designed to operate at very low pressure during the deposition process.

After the coating process, the materials were cross-sectioned accordingly for constant sputtering power and gas flow rate at 200 W and 10 sccm for all four samples, whereas 3 h. with 100 °C for sample A, 2 h. with 80 °C for sample B, 2 h. with 100 °C for sample C, and 3 h. with 80 °C for sample D, respectively.

### 2.2. Thin Films’ Characterisation

The crystalline structures and phase composition of TiC thin films were deposited and characterised by using X-ray diffraction (XRD) from the Bruker diffractometer (Rigaku, Ultima IV). The surface morphology was examined by a scanning electron microscope (SEM) from Zeiss (GEMINI FESEM TECHNOLOGY) and then used to analyse the wear scar and debris after sliding alloy steel balls on the deposited films. The surface morphology and the surface roughness of TiC thin films were observed through the non-contact atomic force microscope (AFM), by using a Veeco Di3100 instrument from Veeco Instruments Inc. The molecular structure of the coatings was analysed by using Fourier transform infrared spectroscopy (FTIR) done in transmission mode with wave numbers in the range of 5000–500 cm^−1^ by the IRAffinity-1 (High-performance FTIR) spectrometer, equipped with a deuterated triglycinesulphate detector. The tribological analysis of the thin films was investigated by using a multi-function tribometer (at room temperature under dry sliding conditions) called the Rtec Instruments (Rtec Instruments, California, USA). An alloy steel ball (diameter of 6.35 mm) was used as the counter body. All the tests were performed at different loads between 20 N and 25 N, respectively. In addition, the optical properties of the thin films along the wear tracks were investigated. However, optical morphology was carried out by using a BX51M optical microscopy instrument. The wear volume of the deposited film was automatically obtained from the Rtec instrument whereas the wear rate was evaluated by using the classical Archard’s wear law [14].

## 3. Results and Discussion

### 3.1. The SEM Micro-Structural Analysis of the Coated TiC Thin Film

The coated TiC thin film on the copper substrate surfaces was investigated by using various equipment for the microstructural behaviour. Thus, the grain structures of the film were fully adhered to on the substrate’s surface, which was characterised by the columnar grain structures. However, Figure 3 shows the micrographs of the fabricated TiC thin film on copper. SEM micrographs of the coated films were observed with densely packed thin film structural grains. Thus, the adverse effect of the sample with the presence of columnar grains could be susceptible to mild wear defects.

### 3.2. Atomic Force Microscopy (AFM) Analysis

The surface roughness for the respective samples varies, due to slight changes in sputtering time and the deposition temperature. However, Table 1 presented the surface roughness and the grain (height) size of various samples observed in this study. Figure 4 presents the topography of the surface roughness for the different samples used in this study.

The TiC thin films were obtained by RF sputtering in the presence of a maintained flow of Argon gas in the sputtering chamber. It can be observed that the surface of sample B films is smoother and more uniform when compared to the films from other samples, which can as well be confirmed from the obtained values of root-mean-square (RMS) roughness. Sample B film has 23.856 nm roughness, which is lower than other samples of film over a 4.175 × 4.175 µm^2^ area. This can be attributed to the free flow of Argon gas and the dispersing effect of Ti, which results in the gentle formation of small amorphous carbon clusters, which in turn prevent relatively large clusters during thin film deposition [16].

### 3.3. XRD Analysis of TiC Thin Film Coatings

Figure 5 shows the XRD patterns of deposited TiC thin films with a peak at 43.7° from the Cu (111) substrate underneath the deposited film. Peaks at 2θ = ~44° and ~51° are attributed to the (111) diffraction plane for Cu(Cu16Ti)_0.23_ and the (200) diffraction plane for Cu(Cu_0.997_Ti_0.003_) with Cu(Cu_0.923_Ti_0.077_) phase identifications of coated thin film on copper substrates. These patterns clearly exhibit relatively broad peaks. The deposited thin films have a dominant orientation of the (200) plane situated at 50.8°, which however belongs to Cu crystallites. The (111) preferential orientation is possibly due to the smallest surface energy storage in the stressed rate. The observed diffraction peaks corresponding to (111); while (200) are visible in all the prepared samples under different parameters. The patterns of coated films obtained from the TiC target exhibit a preferential orientation of the (111) plane. Wang et al. [17] reported that this type of preferential orientation could be attributed to the lowest storage of surface energy in the stressed state. Hence, the increase of mobility of atoms during the deposition process may lead to the relaxation of the film structure to an insufficient energy state, with the formation of (111) texture. The peak at ~32° is a satellite of the major TiC peak caused by CuK_β_ radiation of wavelength 1.39217°A [1].

### 3.4. FTIR Analysis of TiC Thin Film Coatings

The FTIR analysis was performed to investigate the transmission bands and confirm the formation of crystalline TiC thin films. Figure 4 shows the FTIR spectra of TiC films formed under different sputtering times (2–3 h) and temperatures (80 °C–100 °C) on Cu substrates at a constant sputtering power of 200 W in the range of 5000–500 cm^−1^. The peaks at 540, 780, and 1250 cm^−1^ were presented in the FTIR spectra, and the formation of the Ti–C bond was observed. Figure 6 shows the FTIR spectra obtained on the thin films prepared under different times and temperatures. Surmenev et al. [9] reported that the intensities of the FTIR bands vary with the degree of material crystallinity. Foratirad et al. [18] located the spectra region at 466 cm^−1^ and attributed it to the Ti–C vibration. The bands at 664 cm^−1^ and 1250 cm^−1^ in sample D are due to titanium oxide; they are assigned to the lattice vibrations of TiO_2_. The peak centered at 1258 cm^−1^, 1072 cm^−1^, and 1065 cm^−1^ for samples A, B, and C, respectively can be attributed to characteristic O–O stretching vibration. The strong transmission bands at 1497 cm^−1^ are attributed to the vibration of stretching and deformation of the Ti–O–Ti bond lift-up (stemming) from the Cu substrate because of the relatively small size thickness of the thin films. Thus, the transmission bond of 780 cm^−1^ is, however, attributed to the vibration of the Ti–O–Ti bonds in TiO_2_ [10].

### 3.5. Tribological Properties and Analysis

The wear process parameters used for the tribological test are given in Table 2. as well as the wear properties. There is a variation in the load applied, which is given between 20 N and 25 N, respectively. The wear properties are given as follows: Scratch distance S = 3 mm, Velocity = 4 mm/s, Acceleration = 0.1 mm/s^2^, Dwell time (duration) = 0.1 min, and Recipe time = 1 min. The following factors were evaluated at different load applications: Wear volume, V (mm^3^), Average wear depth, L (µm), wear tool position over the specimen, ΔX (µm), wear tool displacement reciprocating on the specimen, ∆H (µm) and the Coefficient of friction (CoF). Thus, force application for different samples is given as follows: Sample A = 20 N, Sample B = 20 N, Sample C = 25 N, and Sample D = 25 N. A ball-on-disc tribometer (MFT Series, Rtec Instruments) was applied to evaluate the tribological properties of the films in an environmental chamber (−100 °C up to 1500 °C and 5 to 95% RH). However, the test counterpart was an alloy steel ball with a diameter, (ᴓ = 6.35 mm).

The load applied the sliding speed, and the sliding distance at 20 N to 25 N, 4 mm/s, and 3 mm, respectively. Hence, the following are the alloy steel ball properties used for the tribological study: Specification is BM110001, Grade 25, and E52100 of Alloy steel. The wear rate *W* of the thin films was evaluated using the proposed Archard’s classical wear equation: [14]. However, Figure 7 illustrates the schematic diagram of wear process for coated TiC thin film on copper substrate.
(1)$$R=\frac{V}{\left(S\right)\times \left(F\right)}$$
where, *R* = Wear rate [mm^3^/Nm], *V* = Wear volume [mm^3^], *S* = Sliding distance [m] and *F* = Applied force [N].

#### 3.5.1. Tribological Analysis for TiC Thin Films Deposited on Copper Substrates

Figure 8 shows the wear track of various samples at different applied loads between 20 N and 25 N, showing the corresponding wear average depth. Thus, sample D has the lowest average wear depth of 723.016 µm; while sample C has the highest average wear depth of 858.229 µm.

#### 3.5.2. Optical Microscopy Analysis

Figure 9a–h shows the optical microscope images of the different behaviours of the tracks after the wear tests. The adhesive failure occurred inside of the wear track for RF film, forming film thinning (Figure 9a,c). The thin film was partially delaminated and thereafter completely removed from the substrate after a certain load value of 25 N, which is shows in Figure 9h. Thus, Figure 9e–f films exhibited higher adhesion performance than other films coated, which could be due to a columnar structure that caused the removal of thin films very quickly [5].

#### 3.5.3. Scanning Electron Microscopy Analysis

Scanning electron microscopy (SEM) tests were conducted to understand the associated mechanisms responsible for the wear resistance in the TiC thin films. The wear tracks formed on the films were investigated by SEM microstructural analysis as shown in Figure 10. Thin films deposited in samples C and D (Figure 10e–h) have narrow wear tracks and little wear debris. On the other hand, more wear debris and slight grooves were detected on the wide wear tracks of the thin films deposited on samples A and B (See Figure 10a–d). Moreover, it is acknowledged that the main wear mechanism of these films shows abrasive wear [14].

#### 3.5.4. Friction and Sliding Wear Behaviour

The friction and dry sliding wear tests conducted on a ball-on-disc tribometer (MFT Series, Rtec instruments) at a constant velocity of 4 mm/s with its associated software were analysed and the Coefficient of friction (CoF) of TiC thin films on copper substrates was investigated. Obtained wear volume was however used to evaluate the respective wear rates, using the classical Archard’s wear equation [14]. The RTEC tribometer software calculates the wear volume and the average wear depth of the alloy steel ball onto the copper substrates and the static and dynamic coefficients of friction. Table 3 presented the average wear depth, wear volume, and wear rate of thin films for different samples. On the other hand, Figure 11 and Figure 12 show the TiC thin film behaviours on Cu substrates for various samples. Figure 11 revealed that the coefficient of friction between samples A and B ranged from 0.21 to 0.23, while the coefficient of friction between samples C and D ranged from 0.33 to 0.38; and was similar in behaviour which corresponds to the load applied.

The load was applied to samples A and B; this was found to be 20 N, while a 25 N load was applied to samples C and D. Thus, samples with a higher coefficient of friction would be more susceptible to film failure. Figure 12a shows the average wear depth behaviour of TiC thin films deposited on the copper substrates. There is a progressive increase from sample A to C, then an observed drastic decrease in sample D. Sample C has the highest average wear depth of 858.229 µm, while sample C was observed with the lowest average wear depth of 723.016 µm at different sputtering times and temperatures, with a constant sputtering power of 200 W.

Furthermore, it was detected that there was an irregular increase-decrease behaviour in both wear volume and wear rate for the respective samples, in which sample C has the highest wear volume of 9.3 × 10^−3^ mm^3^ and the wear rate of 1.24 × 10^−1^ mm^3^/Nm. Both Figure 9b and Figure 9c revealed that sample B was characterised by the lowest wear volume and wear rate of 5.1 × 10^−3^ mm^3^ and 8.48 × 10^−2^ mm^3^/Nm, respectively.

## 4. Conclusions

TiC thin films were successfully formed on copper substrates by using a radio-frequency magnetron sputtering (RFMS) process at a constant sputtering power of 200 W and at varying times and temperatures between 2–3 h and 80 °C–100 °C using Argon (Ar) gas at a flow rate of 10.0 sccm. It has been found that Sample C, which underwent a 2–hour sputtering process at 100 °C, is more susceptible to wear than other samples, where temperature affects wear more so than sputtering time. Thus, the effect of the temperature difference is more profound in the deposition of TiC thin films on copper substrates when compared to the time difference. More ploughs and wear debris are revealed on the wear tracks of the films deposited on sample C, which agrees with both the wear volume and the wear rate results. On the other hand, the effect of deposition time difference revealed lower grain size at an increase in sputtering time when compared to higher grain size at a decrease in sputtering time for constant temperature. Therefore, deposition time provided more effect on grain size than deposition temperature.

The wear is known to be a complicated phenomenon, which is, however, governed by many parameters, such as surface roughness, toughness, and hardness. Thus, the samples with a higher CoF would be more susceptible to film failure. Hence, this paper investigated the effect of sputtered deposition parameters on the microstructural and wear behaviours of thin film titanium carbide coated on metallic copper material via the magnetron sputtering method. Furthermore, the coefficient of friction between samples A and B ranges from 0.21 to 0.23, while the coefficient of friction between samples C and D ranges from 0.33 to 0.38. This is similar in behaviour, which however corresponds to the load applied between 20 N and 25 N.

## Figures and Tables

**Figure 1 materials-16-00174-f001:**
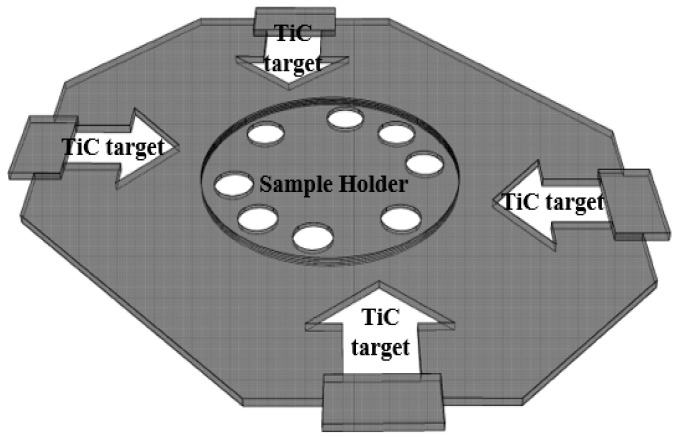
Schematic diagram of RF magnetron sputtering technique.

**Figure 2 materials-16-00174-f002:**
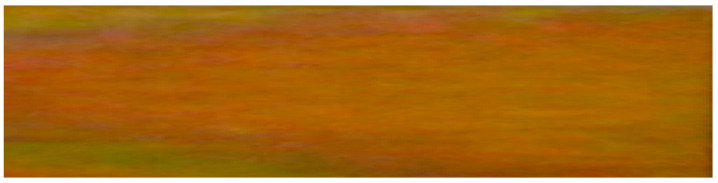
Photographic image of TiC thin film deposited on a copper substrate.

**Figure 3 materials-16-00174-f003:**
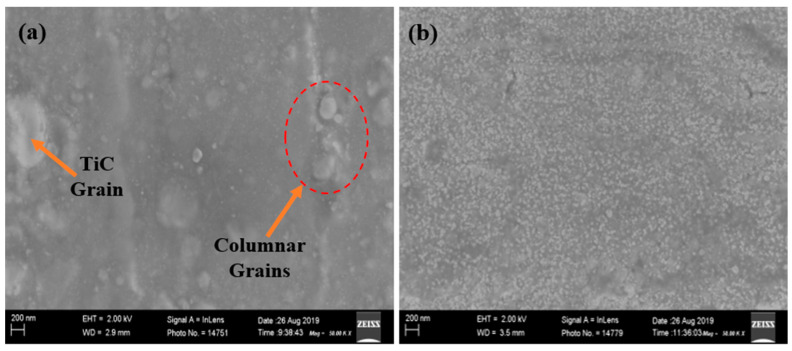
SEM micrographs of TiC thin films coated on copper substrates at X50,000 magnification (**a**) 3 h and 100 °C, and (**b**) 3 h and 80 °C.

**Figure 4 materials-16-00174-f004:**
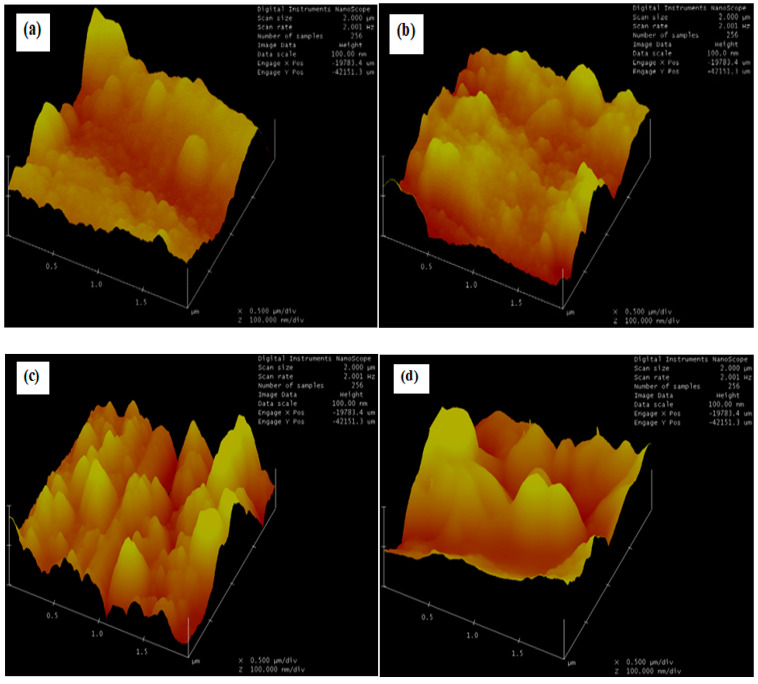
AFM images of sputtered TiC thin films deposited on Cu substrates for different samples (**a**) 3 h and 100 °C, (**b**) 2 h and 80 °C, (**c**) 2 h and 100 °C, and (**d**) 3 h and 80 °C.

**Figure 5 materials-16-00174-f005:**
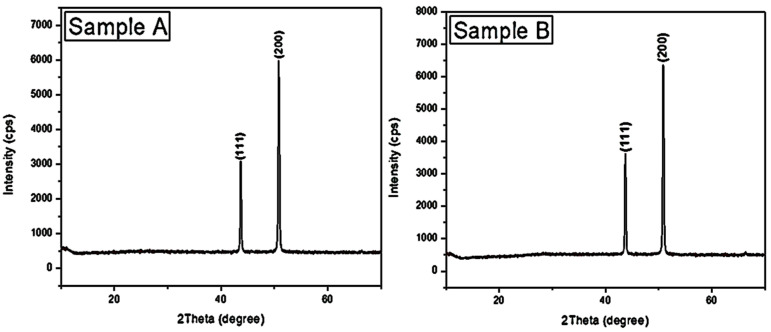
XRD spectra of deposited TiC thin films on the Cu substrates: samples (**A**) 3 h and 100 °C, (**B**) 2 h and 80 °C, (**C**) 2 h and 100 °C, and (**D**) 3 h and 80 °C.

**Figure 6 materials-16-00174-f006:**
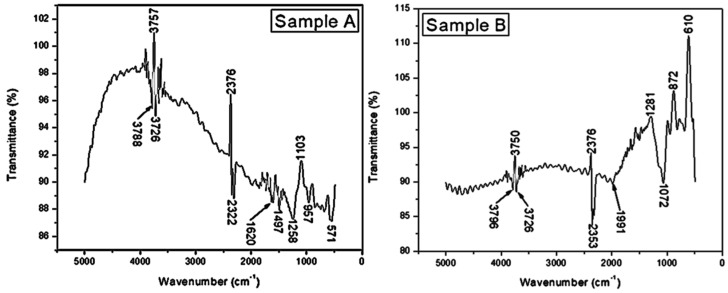
FTIR spectra of deposited TiC thin films on Cu substrates in the wavenumber range of 5000–500 cm^−1^: samples (**A**) 3 h and 100 °C, (**B**) 2 h and 80 °C, (**C**) 2 h and 100 °C, and (**D**) 3 h and 80 °C.

**Figure 7 materials-16-00174-f007:**
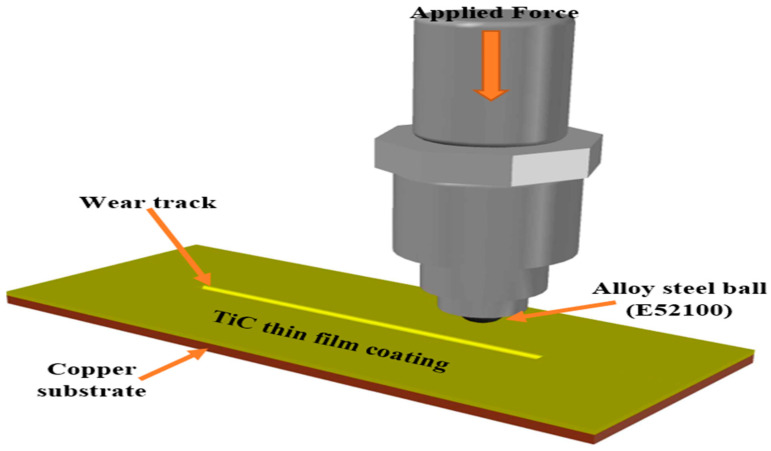
Schematic diagram of wear process for TiC thin film coated on copper substrate.

**Figure 8 materials-16-00174-f008:**
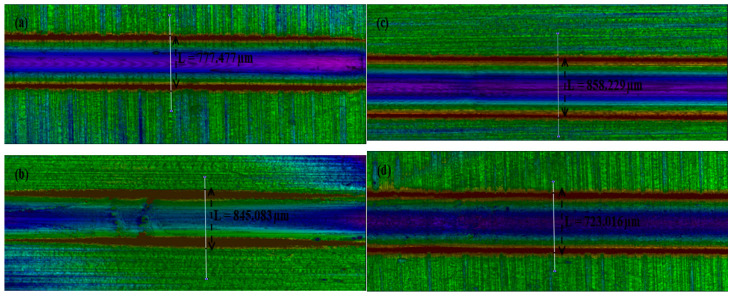
Rtec instrument images for tribological test showing average wear depth at applied load (**a**) 20 N, (**b**) 20 N, (**c**) 25 N, and (**d**) 25 N.

**Figure 9 materials-16-00174-f009:**
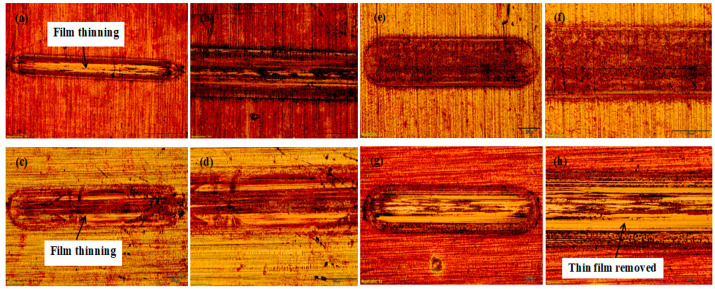
Optical microscopy images showing the surface morphology of films’ wear tracks on a copper substrate.

**Figure 10 materials-16-00174-f010:**
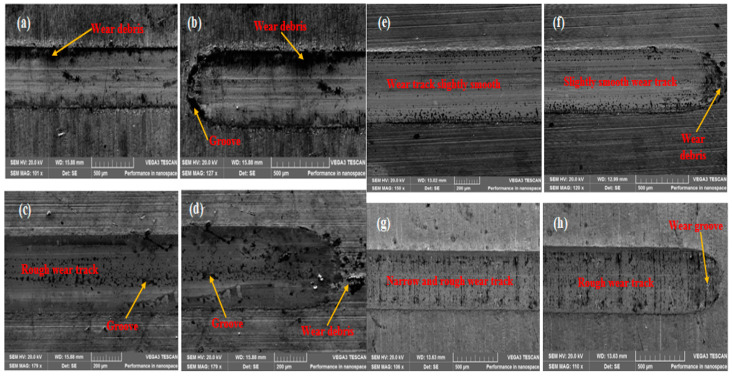
SEM micrographs of the wear track on the coated films.

**Figure 11 materials-16-00174-f011:**
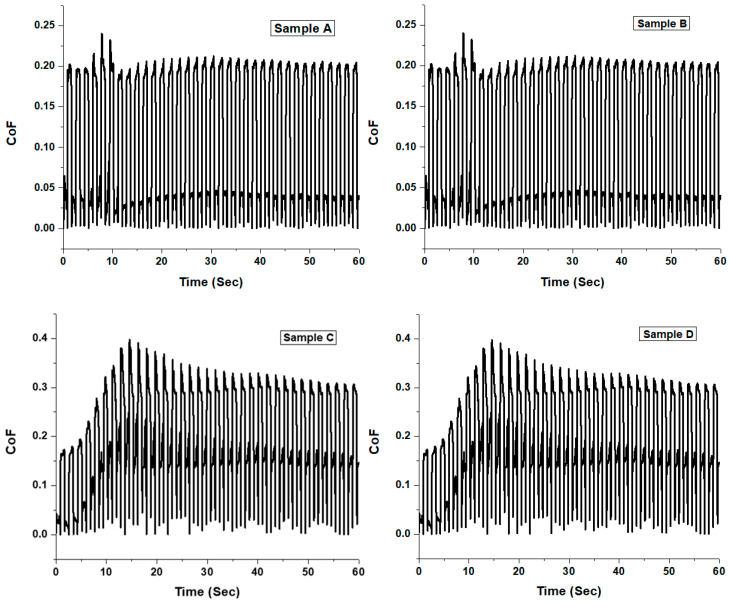
The Coefficient of friction against time for the TiC thin films deposited on the Cu substrates: samples (**A**) 3 h and 100 °C, (**B**) 2 h and 80 °C, (**C**) 2 h and 100 °C, and (**D**) 3 h and 80 °C.

**Figure 12 materials-16-00174-f012:**
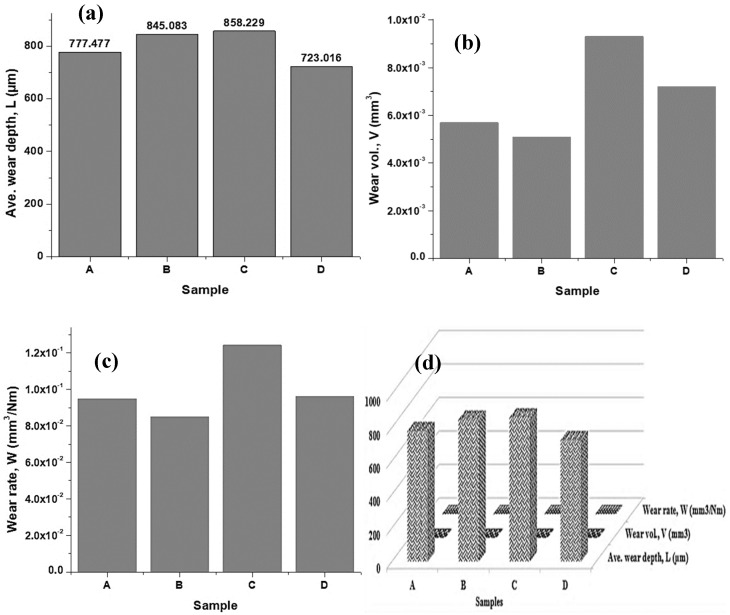
(**a**) Average wear depth of TiC thin films on Cu substrates, (**b**) Wear volume of TiC thin films on Cu substrates, (**c**) Wear rate of TiC thin films deposited on Cu substrates, and (**d**) Combined wear behaviours of TiC thin films on copper substrates.

**Table 1 materials-16-00174-t001:** Data obtained from AFM analysis of TiC thin films deposited on Cu substrates.

Sample	RF Power (W)	Time (h)	Temp (°C)	Argon Gas Flow Rate (sccm)	Grain Size (nm)	RMS (nm)	Roughness, Ra (nm)
A	200	3	100	10	54.240	29.583	24.871
B	200	2	80	10	91.507	23.856	19.797
C	200	2	100	10	140.440	37.029	28.361
D	200	3	80	10	83.152	42.207	33.080

**Table 2 materials-16-00174-t002:** Process parameters for the wear experimental analysis on TiC thin films deposited on Cu substrates.

Sample	Load (N)	L (µm)	∆X (µm)	∆H (µm)	Wear vol. (mm^3^)
A	20	777.477	450.711	0.724	5.692367exp–3
B	20	845.083	490.148	–1.193	5.088833exp–3
C	25	858.229	507.050	0.434	9.310995exp–3
D	25	723.016	482.636	0.143	7.195599exp–3

**Table 3 materials-16-00174-t003:** Sliding wear parameters for TiC thin films on copper substrates.

Sample	Ave. Wear Depth, L (µm)	Wear vol., V (mm^3^)	Wear Rate, W (mm^3^/Nm)
A	777.477	5.692367exp–3	0.094873
B	845.083	5.088833exp–3	0.084818
C	858.229	9.310995exp–3	0.124147
D	723.016	7.195599exp–3	0.095941

## Data Availability

The data presented in this study are available on request from the corresponding author.

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
