# Peer review of "Effect of Microstructural and Tribological Behaviors of Sputtered Titanium Carbide Thin Film on Copper Substrate"

_materials, 2022, doi:10.3390/ma16010174_

Round 1
Reviewer 1 Report
materials-2094954-peer-review-v1
Title:
Effect of microstructural and tribological behaviors of sput-2 tered titanium carbide thin film on copper substrate
Comment:
- Manuscripts need to be read thoroughly in order to avoid typo mistakes as can be seen. For example, in section 2.1, the text is in BOLD format.
- All the abbreviations need to be introduced in the first appearance place.
-Problem statement is not clear.
- The novelty of the job needs to be bold.
- The objective of this work needs to be clearly stated in the last paragraph of the introduction.
-I have read and evaluated the manuscript and in my opinion the submission does not yet sufficiently justify publication. Discuss the shortcomings of previous work and the gaps and how this work intends to fill those gaps. Related references should read and be cited:
Materials Science in Semiconductor Processing, 66 (2017) 157-161, Applied Surface Science, 385 ( 2016)182-190, Journal of Materials Science: Materials in Electronics volume, 28 (2017) 12458–12466, Materials Letters, 210, ( 2018) 4-7, Ceramics International, 44(2018) 18817-18820, Ceramics International, 45 (2019) 20697-20703, Ceramics International, 45 ( 2019)15077-15081
- Fig.1 does not really show the Schematic diagram of the RF magnetron sputtering technique. Need to be redrawn.
- More details need to be added in section 2.1, for example (1) why the author has chosen copper as a substrate, (2) why does working pressure is low for deposition 4.68 x 10-2 Torr (it is supposed to be 10-5), (3) what is the RF deposition power?
- Figure 2 must be removed. The reader can get any info from Fig.2.
- Section 2.2 need to be explained as presented in the results and discussion respectively.
- First paragraph in section 3 “ Surface morphology……”, its well know and must be remove.
- Author need to explain what is the effect of deposition temperature and time on morphology such as grain size.
- In Fig.3, the author put SEM images of (a) 3 hrs & 100 oC, and (b) 3 hrs & 80 oC. How about other deposition temperatures and times?
- The author needs to put an SEM cross-section image to show the thin film thickness changes with deposition temperature and time.
- Table 1, the data need to be presented respectively. rearrange it.
- How the author measured the grain size in table 1, need to be explained.
-The crystallite size of the sample needs to be measured and the effect of deposition temperature and time on Crystallite size should be explained. The author can read and cite the following research work in order to address this comment
Micro & Nano Letters 11 (2016)147-150, Ceramics International, 46 ( 2020) 20313-20319
- The inter-correlation between results is not clear.
-The author needs to make a Table of comparison to show the improvement of present work with others. The author can use the mentioned reference above for comparison as well.
- The reference list is adequate, although in some cases newer or more appropriate publications should have been cited.
If the authors can revise the manuscript based on the suggested points, I will be happy to review this work again.
Author Response
Response to Reviewer 1 Comments
Point 1: Manuscript need to be read thoroughly in order to avoid typo mistakes as can be seen. For example, in section 2.1, the text is in BOLD format.
Response 1: Section 2.1 has never been in BOLD format, however, it has been taken care of.
Point 2: All the abbreviations need to be introduced in the first appearance place.
Response 2: All the abbreviations were mentioned in Abstract and Introduction sections.
Point 3: Problem statement is not clear.
Response 3: This is clearly stated in the manuscript that, the tribological analysis of the titanium carbide (TiC) thin film coated on the metallic copper material has not been discussed by many researchers using radio frequency magnetron sputtering (RFMS) technique.
Point 4: The novelty of the job needs to be bold.
Response 4: The novelty of this research work was clearly stated in the second to the last paragraph (i.e., line 6-8) before the inserted objective paragraph of the Introduction section. However, these materials have not been investigated by many researchers.
Point 5: The objective of this work needs to be clearly stated in the last paragraph of the introduction.
Response 5: The objective of this research study is to investigate the behaviour of TiC thin films deposited on metallic copper plates surface.
Point 6: I have read and evaluated the manuscript and in my opinion the submission does not yet sufficiently justify publication. Discuss the shortcomings of previous work and the gaps and how this work intends to fill those gaps. Related references should be read and be cited.
NB: Some journal articles were however recommended to support this work.
Response 6: I realised that the reviewer only focused attention on microstructural behaviours and not the tribological analysis of this research work. This research study also presented the wear behaviours of TiC thin film coated on copper (base material) substrate at different processing parameters and observed varied average wear depth, wear volume and wear rate, respectively for different samples. Hence, the wear rate was however evaluated using the Archard’s wear equation. Therefore, these parameters could be used for manufacturing industries for cutting tools applications.
Point 7: Fig. 1 does not really show the Schematic diagram of the RF magnetron sputtering technique. Need to be redrawn.
Response 7: Fig. 1 has been redrawn.
Point 8: More details need to be added in section 2.1, for example (1) why the author has chosen copper as a substrate, (2) why does working pressure is low for deposition 4.68 x 10-2 Torr (it is supposed to be 10-5), (3) what is the RF deposition power?
Response 8: Copper material was chosen as a substrate because of its superior chemical and thermal properties. Magnetron sputtering equipment can operate at a very low pressure. The RF deposition power is 200 W and it has been mentioned in the manuscript.
Point 9: Figure 2 must be removed. The reader can’t get any info from Fig. 2.
Response 9: Figure 2 has been adjusted for a better presentation.
Point 10: Section 2.2 need to be explained as presented in the results and discussion respectively.
Response 10: This section mentioned the necessary characterisation methods used to analyse the coated samples. All the explanations have been presented in the results and discussion section of the manuscript. I don’t think there is any further explanation to be made in the section 2.2, thank you.
Point 11: First paragraph in section 3 “Surface morphology…….”, its well known and must be removed.
Response 11: This paragraph has been removed.
Point 12: Author need to explain what is the effect of deposition temperature and time on morphology such as grain size.
Response 12: These effects have been explained in the conclusion section of the manuscript.
Point 13: In Fig. 3, the author put SEM images of (a) 3 hrs & 100 oC, and (b) 3 hrs & 80 oC. How about other deposition temperatures and times?
Response 13: Images not available.
Point 14: The author needs to put a SEM cross-section image to show thin film thickness changes with deposition temperature and time.
Response 14: The cross-section is out of the scope of this work.
Point 15: Table 1, the data need to be presented respectively. Rearrange it.
Response 15: Noted and rearranged.
Point 16: How the author measured the grain size in table 1, need to be explained.
Response 16: Atomic force microscopy (AFM) equipment will automatically measure necessary parameters such as roughness analysis and grain size during operation from the command tools. Therefore, the grain sizes were measured by AFM images.
Point 17: The crystallite size of the sample needs to be measured and the effect of deposition temperature and time on crystallite size should be explained. NB: These journal articles were recommended, “Micro & Nano Letters 11 (2016) 147-150, Ceramics International, 46 (2020) 20313-20319.
Response 17: The measurement of crystallite size is out of the scope of this work. We only considered the grain size.
Point 18: The inter-correlation between results is not clear.
Response 18: Noted
Point 19: The author needs to make a Table of comparison to show the improvement of present work with others.
Response 19: This research work used novel materials (i.e., TiC thin film coated on metallic copper substrate) in which very little researchers have done work on, especially on tribological analysis. Therefore, the results presented in this study could be used in surface manufacturing applications.
Point 20: The reference list is adequate, although in some cases newer or more appropriate publications should have been cited.
Response 20: Noted with thanks.
Remark:
NB: The manuscript has been corrected by English language editor.

Reviewer 2 Report
Manuscript is dedicated to the study of the effect of microstructural and tribological behaviors of sputtered titanium carbide thin film on copper substrate. This paper reports that the coating of TiC thin films on copper can contribute to tribological applications. This area of research is not new, but material contains interesting parts.
In this study, the authors concluded that the temperature affects wear more so than sputtering time. In this viewpoint, experimental data are not well shown; surface roughness, crystalline orientation, wear as a function of time deposition and temperature. However, several points are still unclear. To improve the scope of this paper, I recommend revision on the basis of the following points.
· I advise the authors to rewrite the abstract, because they talk about the samples A, B, C and D and this is not informative for the readers.
· All the references are not well written. There is confusion between the surname and the first name this occurs when the writers use a software like endnote or others. For example reference 9
In the manuscript : Roman, A. S.; Maria, A. S.; Irina, Y. G.; Roman, V. C.; Barbel, K.; Tilo. B.; Kateryna, L.; Matthias, E…. Applied Surface Science. 2017, 414, 335-344.
The right citation:
RA Surmenev, MA Surmeneva, IY Grubova… - Applied Surface …,
· I suggest to the authors to give the first information about the sample’s A, B, C and D in the material and methods.
· The figure 2 is not clear, is it the samples after the deposition or before . If it’s after why the TiC film is transparent? What’s the black lines?
· The introduction required revision , the authors should avoid the repetition. It should be noted briefly why the copper was chosen as the substrate? And what’s the novelty of this study?
· In Table 1, the authors present the value of the roughness up to the third digit after the decimal point, is it physically significant?! By the way, the authors have not mentioned errors in the value of parameters throughout the manuscript.
Now I come to scientific part of the manuscripts. Authors have investigated, surface morphology, crystal structure and tribological properties of RF magnetron sputtered TiC thin films. My main critics are as follows:
· The authors state that the grain size was obtained by Atomic force microscopy (AFM) , the images are not clear to calculate the grain size so it’s better to use the XRD for this matter.
· In the manuscript the Figure 3 shows the micrographs (observed by SEM) of the deposited TiC thin film on copper for two temperatures is it possible to add the two other deposition time. Also , the authors did not make any comment on its effect on different morphologies of the growing film.
· Figure 4 is not clear.
· For the results of XRD , it’s not clear how the phases like Cu(Cu16Ti)0.23 , Cu(Cu0.997Ti0.003) and Cu(Cu0.923Ti0.077) were obtained . In my opinion, this identification is not correct . The peaks are clearly attributed to TiC. All such things need to be addressed.
· The XRD patterns displays two peaks , it seems that there is a small shift of the (111)
· I suggest to change the word “fabricated” by “deposited” or any other word in this field of research
· The Schematic diagram of RF magnetron sputtering technique should be improved because the pink color seems like bulk material
· Figure1, figure 2, Figure 4 , figure 5 , figure 6 figure 8 , figure 10 and figure12 should be improved
Author Response
Response to Reviewer 2 Comments
Point 1: I advise the authors to rewrite the abstract, because they talk about the samples A, B, C and D and this is not informative for the readers.
Response 1: Noted.
Point 2: All the references are not well written. There is confusion between the surname and the first name, this occurs when the writers use a software like endnote or others. For example, reference 9.
Response 2: Noted and fixed.
Point 3: I suggest to the authors to give the first information about the sample’s A, B, C and D in the material and methods.
Response 3: First information has been provided for all the samples as suggested.
Point 4: The figure 2 is not clear, is it the samples after the deposition or before. If it’s after, why the TiC film is transparent? What’s the black lines?
Response 4: The figure 2 shows the coated TiC thin film after the deposition process. It is a thin film, so the film was formed looking like that. And the black lines on the coated surface were provided for cutting the materials into an appropriate size for microstructure and wear tests.
Point 5: The introduction required revision, the authors should avoid the repetition. It should be noted briefly why the copper was chosen as the substrate? And what’s the novelty of this study?
Response 5: The novelty of this research work was clearly stated in the second to the last paragraph (i.e., line 6-8) before the inserted objective paragraph of the Introduction section. However, these materials have not been investigated by many researchers.
Point 6: In Table 1, the authors present the value of the roughness up to the third digit after the decimal point, is it physically significant? By the way, the authors have not mentioned errors in the value of parameters throughout the manuscript. Now I come to scientific part of the manuscript. Authors have investigated, surface morphology, crystal structure and tribological properties of RF magnetron sputtered TiC thin films. My main critics are as follows:
Response 6: AFM images were captured with 3 decimal places and the values are physically significant.
Point 7: The authors state that the grain size was obtained by Atomic force microscopy (AFM), the images are not clear to calculate the grain size so it’s better to use the XRD for this matter.
Response 7: The grain size values were obtained automatically from the AFM Grain size image analysis, and it can be given if requested.
Point 8: In the manuscript the Figure 3 shows the micrographs (observed by SEM) of the deposited TiC thin film on copper for two temperatures is it possible to add the two other deposition time. Also, the authors did not make any comment on its effect on different morphologies of the growing film.
Response 8: The other two images are not available. Also, comments were made on the effect in the first paragraph, line 6 of section 3.1 of the manuscript.
Point 9: Figure 4 is not clear.
Response 9: Figure 4 is AFM images. That is the best resolution obtained for the manuscript.
Point 10: For the results of XRD, it’s not clear how the phases like Cu(Cu16Ti)0.23, Cu(Cu0.99Ti0.003) and Cu(Cu0.923Ti0.077) were obtained. In my opinion, this identification is not correct. The peaks are clearly attributed to TiC. All such things need to be addressed.
Response 10: Those were the phases obtained by XRD spectra and have been explained in the manuscript with some references.
Point 11: The XRD patterns displays two peaks, it seems that there is a small shift to the (111).
Response 11: Noted.
Point 12: I suggest to change the word “fabricated” by “deposited” or any other word in this field of research.
Response 12: Noted with thanks.
Point 13: The schematic diagram of RF magnetron sputtering technique should be improved because the pink color seems like bulk material
Response 13: Noted. It has been replaced.
Point 14: Figure 1, figure 2, figure 4, figure 5, figure 6, figure 8, figure 10 and figure 12 should be improved.
Response 14: Noted with thanks.
Remark:
NB: The manuscript has been corrected by English language editor.

Round 2
Reviewer 1 Report
The author has addressed all the comments satisfactorily. However, I would recommend the manuscript for publication in this form.